

# Driving factors of runoff and sediment yield effects of various slope ecological restoration measures in the middle reaches of the Yellow River

Xin Wang[1,2,3], Zhenqi Yang[1,2,3], Jianying Guo[1,2,3], Fucang Qin[1,4], Xinyu Liu[2,3] and Dong Fan[1]

[1] Inner Mongolia Agricultural University, Hohhot, China
[2] Ministry of Water Resources Pastoral Area Water Conservancy Science Research Institute, Hohhot, China
[3] Yinshanbeilu National Field Research Station of Steppe Eco-Hydrological System, Hohhot, China
[4] Inner Mongolia Academy of Forestry Sciences, Hohhot, China

Corresponding authors
Zhenqi Yang, yangzq@iwhr.com
Jianying Guo, guojy@iwhr.com

## ABSTRACT

The middle reaches of the Yellow River, with an exceeding amount of coarse sediment compared to the stream flow and the lower reaches, with severe sediment deposition, are key regions for sediment control in the Yellow River Basin. Recent years have witnessed efforts to return farmland land to forest and grassland and the launch of the Three-North Shelterbelt Forest Program, but the effectiveness of these measures remains to be studied. Research on factors influencing runoff depth (RD) and sediment yield (SY) sheds light on the mechanism of soil erosion in the study area. The present study focuses on the standard runoff fields in the Kuye River Basin, where ecological restoration measures (arbor forest land, shrub grassland, natural grassland, artificial grassland, cultivated land, Bare land) for various slope steepness (S) have been taken. Based on a six-year observation of the SY and RD in these fields, we aim to identify the primary factors influencing soil erosion, based on rainfall data and slope gradients. Using rainfall data and slope steepness factors, we explored the dominant factors that influence runoff and SY. The results showed that: (1) the rainfall events with short-duration, medium rainfall, and medium rainfall intensity were the most frequent; (2) the rainfall events with medium duration, heavy rainfall, and heavy rainfall intensity produced the most serious runoff and sediment; (3) using machine learning methods, the researchers found that the gradient boosting decision tree (GBDT) model was the most suitable for the study area, as it provided the best simulation of soil erosion. The structural equation model reveals that there is a significant correlation between runoff depth (RD) and soil erosion modulus (SEM). Time of precipitation (T), average precipitation intensity ($I_{avg}$), maximum intensity of precipitation in thirty minutes ($I_{30}$) and slope steepness (S) are factors that indirectly influence runoff SY. The present study provides technical guidance for the ecological restoration and improvement of different slope surfaces in major sections of the middle reaches of the Yellow River.

## INTRODUCTION

Restoring the Yellow River Basin and managing its sustainable development is one of China's major development projects (*Zuo et al., 2023*). The frequent occurrence of soil erosion disasters in the middle reaches of the Yellow River profoundly impacts the river's sediment discharge (*Mbezele et al., 2022*). *Liu et al. (2024b)* used analytical tools such as the Mann–Kendall (M–K) trend test and double mass curve analysis to examine erosion patterns in these rivers from 1950 to 2022, as well as the primary drivers of these changes. *Wang & Sun (2021)* pointed out that the middle reaches contribute over 90% of the sediment deposited in the lower reaches of the Yellow River and 30% of its annual river discharge. Coarse sediment causes the uncoordinated relationship between runoff and sediment in the Yellow River, which makes the local ecological environment fragile. Therefore, it is important to study the relationship between runoff and sediment (*Wang et al., 2022*). The Miaochuan Basin of the middle reaches of the Yellow River, located in the Mu Us Sandy Land, is an arsenopyrite sandstone area that suffers from severe soil erosion due to the soil type (*Yuan et al., 2025*). China is highly concerned about the ecological and environmental issues in the middle reaches of the Yellow River. Since the 1980s, it has launched a series of projects for ecological conservation, such as the Three-North Shelterbelt Forest Program, the Project for Returning Farmland to Forests and Grassland and the Project for Side Slope Protection. These initiatives have significantly contributed to the prevention and control of soil erosion in the middle reaches of the Yellow River (*Liu et al., 2025*). However, the mechanism of runoff and sediment yield under various vegetation ecological restoration measures in this area remains unclear. *Yin et al. (2024)* showed that runoff depth (RD) and sediment yield (SY) are mainly influenced by precipitation, vegetation, landform and ecological restoration measures. The influence of rainfall on the water erosion process has great spatial and temporal heterogeneity, which is related to the characteristic factors such as rainfall, rainfall intensity, and rainfall duration (T) (*Wu et al., 2017*). *Wischmeier & Smith (1958)* pointed out that the product of rainfall kinetic energy (E) and maximum intensity of precipitation in thirty minutes ($I_{30}$) serves as a measure of precipitation erosivity. This erosivity index is used as an input parameter in the development of the Universal Soil Loss Equation (USLE), a model for predicting soil erosion caused by rainfall. According to *Bochet, Poesen & Rubio (2018)*, there are complex relations between precipitation and runoff and both runoff depth and sediment yield are subject to precipitation intensity. *Hamed et al. (2002)* and *Martinez-Murillo et al. (2013)* found that runoff depth and sediment yield change with precipitation intensity, when other conditions remain relatively constant. Vegetation and topography have strong regional characteristics. Vegetation has the effects of increasing surface coverage, reducing flood peaks, and fixing soil (*Li et al., 2016*). Slope steepness (S) and slope length are two factors related to the implications of landform on soil erosion (*Liu, Zhang & Yun, 2002*). Slope length, slope steepness and slope aspect, which are three main features of a slope, affect not only the manner but also the amount of erosion. Slope steepness has a complex relationship with the process. It can increase the downward gravitational force of rainwater, thus increasing the runoff velocity, reducing the cumulative infiltration, and increasing

the runoff sediment (*Li et al., 2016*). The ecological restoration measures have notable effects on soil erosion, which is greatly affected by human activities and has various degrees of influence on erosion (*Wei & Jiao, 2021*). These factors increase the complexity of the runoff and sediment yield mechanism and also increase the uncertainty of soil water erosion process. Compared to traditional soil erosion models, the machine learning model is able to predict erosion more promptly and efficiently (*Liu et al., 2024c*). *Rahmati et al. (2017)* employed multiple models to predict the soil erosion across lands for different purposes in Iran. Many Chinese researchers (*e.g.*, *Liu, Fan & Wang, 2024*) have also studied the sediment yield of soil erosion in different regions using the machine learning model. Despite years of research, the Mu Us Sandy Land still lacks a machine learning-based model capable of predicting sediment yield from erosion and validating the prediction. Instead, remote sense imaging has been commonly used for simulation and prediction in this region, but its accuracy needs to be improved.

The present study aims to reveal the sediment yield from erosion in a quantitative approach with the machine learning model, according to the data collected by the soil and water conservation monitoring station in Miaochuan Basin from 2018 to 2023. Based on the basic features of runoff sediment, the researchers aim to achieve the following two objectives: 1. to select suitable machine learning models for the study area to simulate and validate the RD and SY under different ecological restoration measures; 2. to identify factors that directly or indirectly affect RD and runoff SY. The results provide theoretical guidance to the comprehensive prevention and control of soil erosion in the sand-covered pisha sandstone area.

## OVERVIEW OF THE STUDY AREA

The Kuye River Basin has the most serious sediment yield problem in the Yellow River (*Zhao et al., 2019*). The upper reaches of the Kuye River Basin are characterized by a landscape covered with sand dunes and shifting sands, while the lower reaches are located in a loess hilly and gully region. The natural ecosystem of the basin is fragile and the soil erosion is serious. It is also affected by human activities such as mineral exploitation and urban construction, which leads to structural and functional issues in the basin ecosystem, the obstruction of the hydrological cycle, and water environment and mine restoration problems (*Zhao et al., 2019*). The most typical sediment yield in the Kuye River Basin results from gravity erosion caused by rainstorm floods, and the maximum recorded sediment concentration is 1,700 kg/m$^3$. Since 1980, the maximum sediment concentration of Wenjiachuan station is 1,420 kg m$^3$, the maximum sediment discharge is 10.002 million tons, and the maximum sediment transport modulus is 1,157 tons km$^2$ (*Wang et al., 2022*). The study area was located on the right bank of the upper reaches of the four-level tributary of the Yellow River Basin, Ordos (109°31′30.97″E, 39°39′2.89″N) (Fig. 1). Situated at the northern edge of the Mu Us Sandy Land, the area is characterized by sand-covered hills and gullies. The topsoil of the region is loose and of no structure or a single-grain structure, with low water and nutrient retention capacity. The main types of soil include calcaric cambisol, aeolian sandy soil, pisha sandstone, and sand-covered pisha sandstone. Constructed in

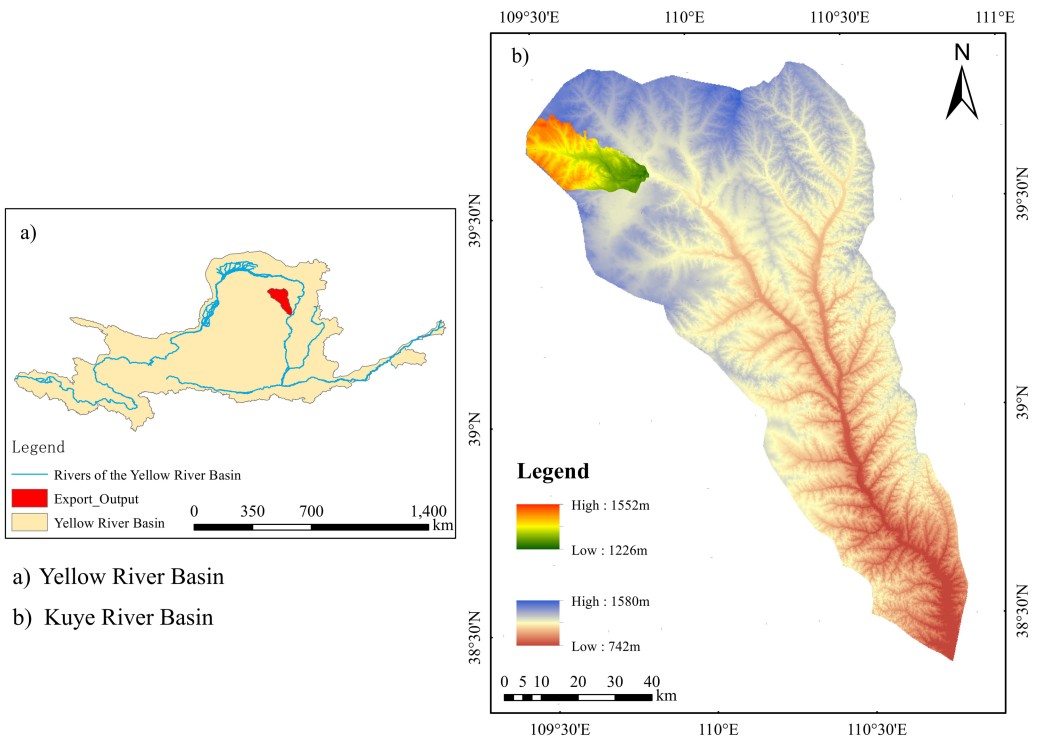

**Figure 1  Overview of the study area.**

2016, Hetongmiao Soil and Water Conservation Monitoring Station (see Fig. 2), Ejin Horo Banner, Ordos, was selected as the study area. The Miaochuan Basin located within the study area belongs to an arid and semiarid temperate continental climate. The average, maximum, and minimum annual rainfall are 358.2 mm, 642.7 mm, and 100.8 mm, respectively, and the average annual sunshine hours are 2,900. The effective accumulated temperature of $\geq 10\,°C$ is 2,751.3 °C, and the average annual evaporation is 2,563 mm. The northwest wind dominates the area. The wind force is 5–8, the average annual wind speed is 3.6 m/s, and the maximum instantaneous wind speed is 24 m/s.

## MATERIALS & METHODS

Three groups of standard runoff plots (5°, 10°, and 15°) were set up for runoff and sediment monitoring in the comprehensive monitoring station of soil and water conservation. The slope length of the runoff plot was 20 m, and the width was 5 m. Each group of plots was set up with *Pinus sylvestris*, *Hippophae rhamnoides*, artificial grassland, crops, natural grassland, and bare land. These plots of various surface forms basically represent the surface conditions of aeolian sandy soil, Pisha sandstone, and the hilly region of the Loess Plateau, and have a wide range of representative significance. The basic overview of runoff field ecological restoration measures (arbor forest land, shrub grassland, natural grassland, artificial grassland, cultivated-land, bare land) is shown in Table 1.

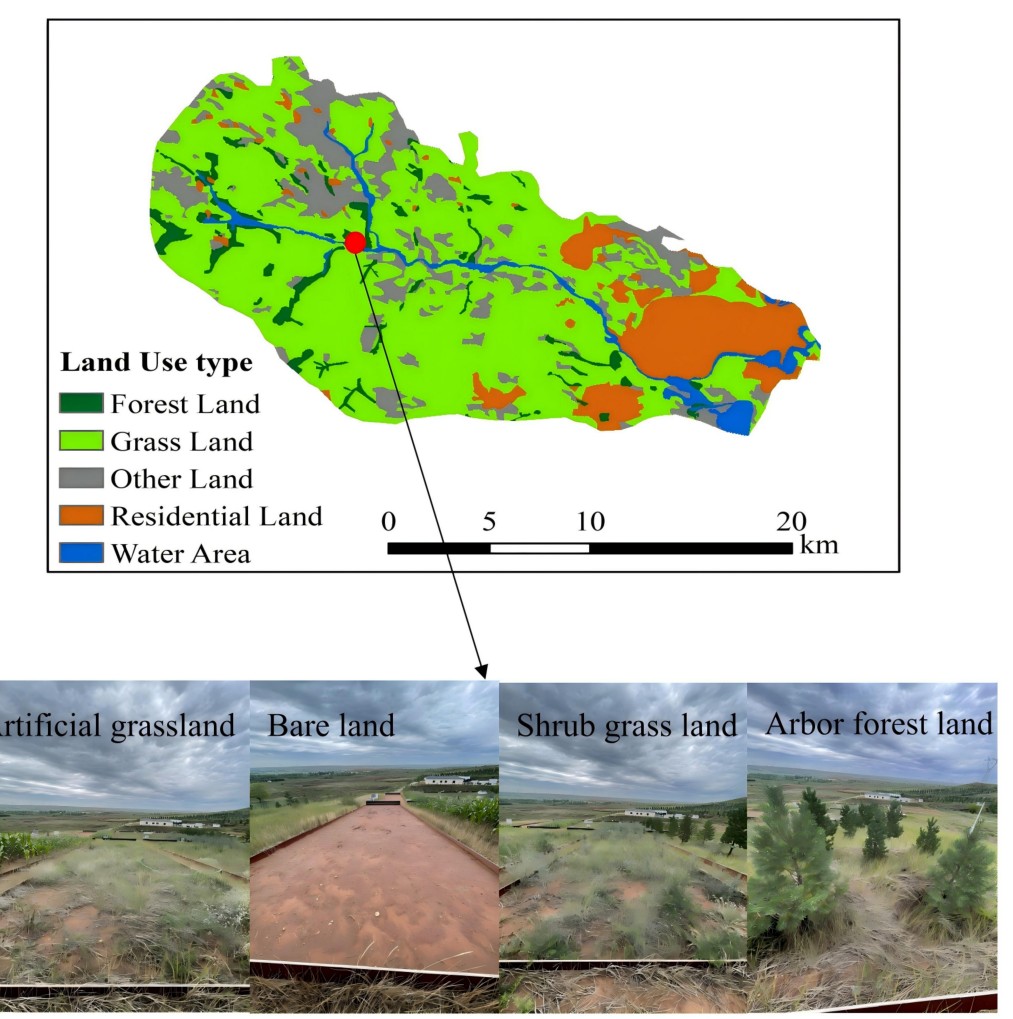

**Figure 2** Overview map of various treatment measures after rainfall.

**Table 1** Basic overview of runoff field treatment measures.

| Configuration mode | Arbor forest land | Shrub grass land | Natural grassland | Artificial grassland | Cultivated-land | Bare land |
|---|---|---|---|---|---|---|
| major vegetation | *Pinus sylvestris* | *Hippophae rhamnoides* | – | *Medicago sativa* | maize | – |
| sowing method | Fish-scale pit | Hole seeding | – | Broadcast sowing technique | Drill seeding | – |
| distance between plants (m×m) | 2×3 | 1×1 | – | – | – | – |
| vegetation coverage% | 46.33 ± 1.25 | 34 ± 1.63 | 43.67 ± 2.62 | 13.33 ± 1.25 | – | – |
| Plot size (m × m) | 20×5 | 20×5 | 20×5 | 20×5 | 20×5 | 20×5 |

(1) Observation of rainfall characteristics.

The rainfall process was recorded by the weather station and observed every 5 min. The rainfall was accurate to 0.1 mm and the rainfall intensity was unit mm/min, combined with runoff and sediment yield (SY) on S under natural rainfall conditions in this area.

**Table 2  Characteristics of various rainfall types.**

| Rainfall pattern | P/mm | | | T/h | | | $I_{30}$max/(mm h-1) | | | $I_{avg}$/(mm h-1) | | | Frequency |
|---|---|---|---|---|---|---|---|---|---|---|---|---|---|
| | Max | Min | Average | Max | Min | Average | Max | Min | Average | Max | Min | Average | |
| I | 72.4 | 0.2 | 6.97 | 510 | 5 | 153.08 | 67.04 | 0.41 | 8.26 | 17.36 | 0.02 | 1.58 | 199 |
| II | 69.4 | 0.8 | 16.55 | 1,560 | 545 | 893.26 | 59.62 | 0.41 | 10.09 | 15.61 | 0.08 | 2.57 | 46 |
| III | 8.4 | 3.8 | 6.1 | 3,000 | 2,530 | 2,765 | 0.82 | 0.41 | 0.61 | 0.87 | 0.39 | 0.63 | 2 |

**Notes.**
I: Short duration, moderate rainfall, moderate rainfall intensity.
II: Medium duration, heavy rainfall, heavy rainfall intensity.
III: Long duration, small rainfall, small rainfall intensity.

For each rainfall event, the rainfall (P), rainfall duration (T), maximum 30-min rainfall intensity ($I_{30}$), and average rainfall intensity ($I_{avg}$) were selected, and the rainfall types in the study area were divided using the system clustering method. The characteristics of rainfall in the study area in 2018-2023 is shown in Table 2.

(2) Runoff sediment observation.

Using automatic water and sediment monitoring equipment (Beijing Tianhang Jiade, Beijing, China), the runoff volume was recorded once a minute, and the runoff volume was accurate to 0.001 L. The sediment yield was accurate to 0.001 kg/m$^3$, supplemented by full-section samplers for manual field sampling. Runoff depth and sediment yield were calculated using the following formulas:

Runoff depth (RD) (mm) = Runoff volume (L)/Plot area (m$^2$)

SY (t/hm$^2$) = Total sediment (t)/Plot area (hm$^2$).

(3) Vegetation observation.

The plant diversity, vegetation coverage, and height and quantity of 1 m × 1 m grass grid quadrats in each vegetation type plot were measured using the photographic method and visual estimation method from April to October, once every 15 days. The plant height, crown width, diameter at breast height, coverage, and spacing of trees and shrubs were measured by ruler from April to October. After the runoff of the plot, the canopy density of trees, coverage of shrub and grass crops, and ground coverage were measured.

(4) The machine learning model is employed.

The present study attempts to predict soil erosion with the neural network (NN), support vector regression (SVR), gradient boosting decision tree (GBDT), and K-nearest neighbors (KNN) and compare the performance of these models. The neural network is a representative of connectionist algorithms, the neuron being its smallest functioning unit. After the weighting of information X input into a neuron from the previous neuron, it is turned into information Y by a response function (*e.g.*, sigmoid function). The foundation of the tree model is the decision tree. This method imitates the way humans make decisions, with the binary tree structure, such as yes/no and true/false (*Breiman, 2001*). Through data examination and evaluation, the scope of the answer is narrowed down. The support vector machine is a machine learning method based on the statistical learning theory. This model, characterized by high generalizability, strikes a balance between model complexity and learning ability with limited sample information.

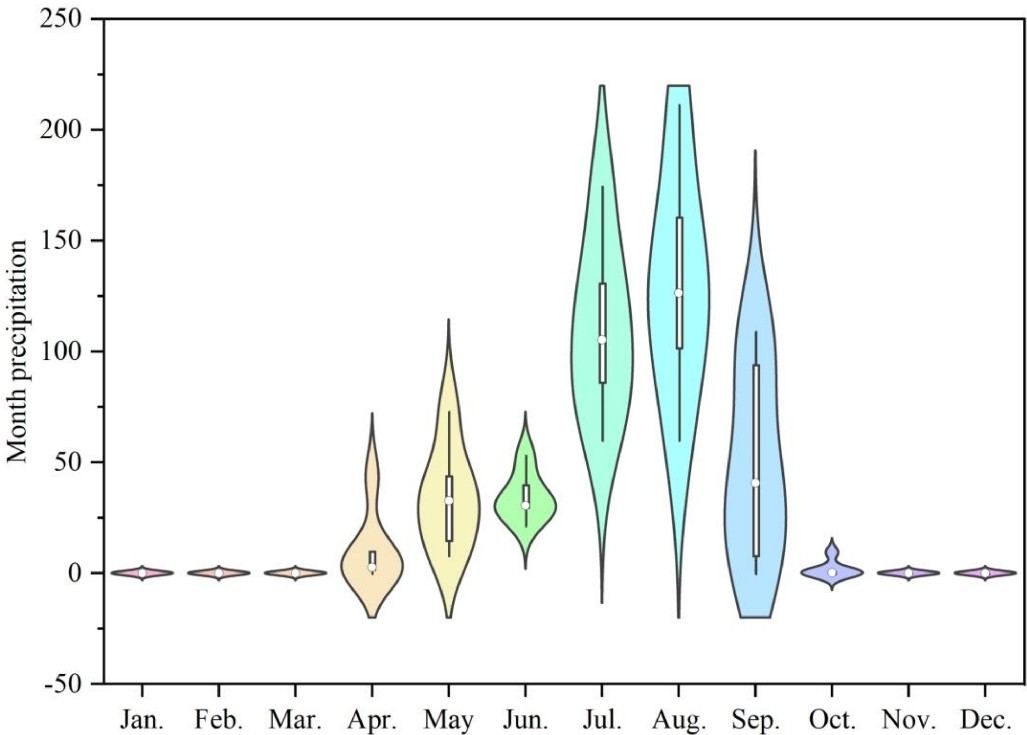

**Figure 3** **The 12-month rainfall distribution map of the study area.**

(5) Statistical analysis.

The experimental data were collated and counted using Excel and plotted by Origin2018. SPASS26.0 was used for two-way analysis of variance (ANOVA), cluster analysis, and correlation analysis to study the connection between water and sediment under various ecological restoration measures. SPSS26.0 was used for path analysis, and the structural equation model was used for regression analysis. The machine learning model utilizes R software for simulation and validation.

## RESULTS AND ANALYSIS

### Rainfall characteristics

A total of 247 rainfall events were observed in the study area from 2018 to 2023. According to the monthly rainfall distribution characteristics, we found that the rainfall in this area is mainly concentrated in July–September (Fig. 3). The rainfall types were further divided by cluster analysis of rainfall characteristics (Table 2). The observed rainfall can be divided into three categories. Type I rainfall occurred 199 times, and was characterized by short duration, moderate rainfall, and moderate rainfall intensity. Type II rainfall occurred 46 times and was characterized by medium duration, heavy rainfall, and heavy rainfall intensity. Type III was characterized by long duration, small rainfall, and small rainfall intensity. The percentages of type I, type II, and type III rainfalls were 80.6%, 18.6%, and 0.8%, respectively. Table 2 indicates that the maximum $I_{avg}$ decreased in the order: type

**Table 3  Basic situation of rainfall in the study area from 2018 to 2023.**

| Year | Annual rainfall/mm | Duration of annual rainfall/h | Maximum rainfall/mm | Maximum $I_{30}$/(mm h$^{-1}$) | Maximum rainfall erosivity (MJ m hm$^{-2}$ h$^{-1}$) | (I + II)/% |
|------|------|------|------|------|------|------|
| 2018 | 388 | 8,145 | 58.5 | 26 | 136.94 | 27 |
| 2019 | 208.6 | 23,460 | 27.6 | 13.49 | 56.16 | 47 |
| 2020 | 363.2 | 12,810 | 44.6 | 62.55 | 766.91 | 49 |
| 2021 | 367.2 | 11,470 | 69.4 | 67.04 | 819.27 | 40 |
| 2022 | 456 | 10,090 | 72.4 | 55.24 | 929.81 | 38 |
| 2023 | 438.6 | 11,107 | 59 | 59.62 | 1,092.75 | 46 |

**Notes.**
(I + II) represents the total proportion of type I and type II rainfall times.

I > type II > type III. The study also found that the rainfall in 2019 was the least (208.6 mm), but the T was the longest and the $I_{30}$ was the smallest (Table 3). The T in 2018 was noticeably less than that in 2019, while the rainfall and $I_{30}$ were higher than those in 2019, indicating that the rainfall was mainly affected by rainfall intensity.

## Runoff and sediment yield response to rainfall patterns under various ecological restoration measures

Across various rainfall types, there was a small variety of responses of RD or SY to S (Fig. 4). The RD and SY of bare land on each slope steepness were noticeably greater than other ecological restoration measures except for crop land. Overall, the steeper the slope, the greater the runoff depth and soil loss. On a 15-degree slope, which is considered steep, the erosive power of runoff increases. Accordingly, the effectiveness of soil and water conservation measures varies significantly. The Type III precipitation causes the shallowest runoff and lowest soil loss. Through correlation analysis of rainfall process, we found a correlation between RD, SY, T, P, $I_{avg}$, $I_{30,figS}$ (Table 4). There was a noticeable ($p < 0.05$) positive correlation between P, RD, and SY of various ecological restoration measures. The effects of T, $I_{30}$, and $I_{avg}$ on runoff and sediment were not strongly correlated under various ecological restoration measures. S showed a noticeable ($p < 0.05$) or extremely noticeable ($p < 0.01$) positive correlation with RD in arbor forest land, natural grassland, cultivated land, and bare land. Table 5 reveals that there is a significant correlation between the RD and SY under various ecological restoration measures.

## Validation of predictions of RD and SY on lands of different uses based on the machine learning model
### The RD model
The data are randomly split into a training set (75%) and a test set (25%), and NN, SVR, GBDT and KNN are trained on RD. Then, the hyperparameters of these models are adjusted using grid search and the accuracy of each model is validated based on the test set (Table 6). $R^2$ can reflect a model's accuracy of prediction. The higher this value, the higher the accuracy of prediction. GBDT outperforms other models in terms of accuracy. The $R^2$s of forest land, shrubland, artificial grassland, natural grassland, farmland, and bare land are 0.92, 0.90, 0.92, 0.91, 0.96 and 0.97 respectively. The $R^2$s of other models are mostly

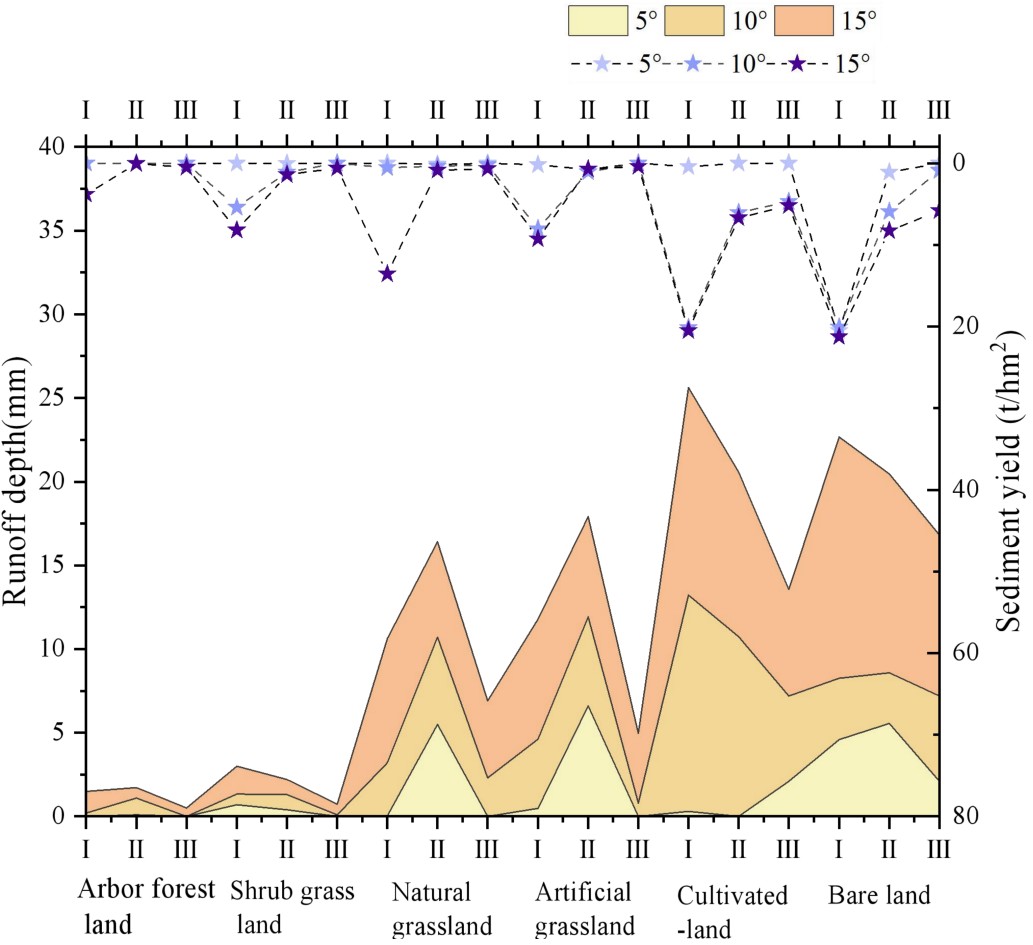

**Figure 4 Characteristics of runoff depth and sediment yield under various slope gradients and various control measures.**

0.32 or above. KNN has the lowest $R^2$. The lower the mean squared error (MSE)—the average of the squared residuals—the smaller the prediction error. The MSEs of forest land, shrubland, artificial grassland, natural grassland, farmland, and bare land under the GBDT model are 0.09, 0.01, 1.94, 1.94, 3.88 and 3.15, respectively. The prediction accuracy of the remaining models is below 16.754. KNN has the lowest MSE. Therefore, KNN is not suitable for the study zone, and GBDT is the most accurate in predicting RD among the four models.

### The SY model

The data are randomly split into a training set (75%) and a test set (25%), and NN, SVR, GBDT and KNN are trained on SY. Then, the hyperparameters of these models are adjusted using grid search and the accuracy of each model is validated based on the test set. Finally, average test results are obtained using the optimal set of hyper parameters (Table 7). In terms of accuracy, GBDT outperforms other models in the simulation of SY. The $R^2$s of forest land, shrub grassland, artificial grassland, natural grassland, farmland, and bare

**Table 4  Correlation analysis of influencing factors of runoff and sediment yield under various control measures.**

| Control measures | Parameter | P | T | $I_{30}$ | $I_{avg}$ | S |
|---|---|---|---|---|---|---|
| Arbor forest land | RD | .538* | 0.169 | 0.144 | 0.212 | 0.999** |
|  | SY | .750** | 0.013 | 0.406 | 0.217 | 0.504* |
| Shrub grassland | RD | .762** | 0.192 | 0.367 | 0.052 | 0.281 |
|  | SY | .611** | 0.033 | 0.357 | 0.151 | 0.294 |
| Natural grassland | RD | .510* | 0.125 | 0.115 | 0.123 | .576* |
|  | SY | 0.449* | 0.017 | 0.243 | 0.118 | 0.315 |
| Artificial grassland | RD | .575* | 0.392 | 0.143 | 0.141 | 0.362 |
|  | SY | .652** | 0.038 | 0.35 | 0.16 | 0.342 |
| Cultivated-land | RD | 0.494* | 0.086 | 0.164 | 0.093 | .717** |
|  | SY | 0.557* | 0.018 | 0.263 | 0.163 | 0.426* |
| Bare land | RD | .530* | 0.092 | 0.145 | 0.118 | .498* |
|  | SY | .588* | 0.008 | 0.325 | 0.211 | 0.816** |

Notes.

Asterisks indicate whether there is a significant difference between factors. One asterisk indicates $P < 0.05$, and two asterisks indicate $P < 0.01$.

**Table 5  Correlation analysis of runoff and sediment under various control measures.**

| Parameter | RD | | | | | |
|---|---|---|---|---|---|---|
|  | Arbor forest land | Shrub grass land | Natural grassland | Artificial grassland | Cultivated-land | Bare land |
| SY | .841** | .788** | .787** | .722** | .894** | .666** |

Notes.

Asterisks indicate whether there is a significant difference between factors. Two asterisks indicate $P < 0.01$.

**Table 6  Validation of the predicted results from the RD models of different control measures.**

|  | NN model | | SVR model | | GBDT model | | KNN model | |
|---|---|---|---|---|---|---|---|---|
|  | $R^2$ | MSE | $R^2$ | MSE | $R^2$ | MSE | $R^2$ | MSE |
| Arbor forest land | 0.81 | 1.08 | 0.83 | 1.01 | 0.92 | 0.09 | 0.64 | 0.28 |
| Shrub grass land | 0.89 | 0.02 | 0.71 | 1.94 | 0.90 | 3.88 | 0.60 | 0.30 |
| Natural grassland | 0.56 | 8.26 | 0.79 | 4.31 | 0.88 | 2.09 | 0.47 | 11.54 |
| Artificial grassland | 0.48 | 9.71 | 0.89 | 2.13 | 0.92 | 1.94 | 0.32 | 12.35 |
| Cultivated land | 0.95 | 6.6 | 0.89 | 10.53 | 0.96 | 3.88 | 0.87 | 12.48 |
| Bare land | 0.90 | 4.96 | 0.82 | 8.78 | 0.97 | 3.15 | 0.64 | 16.75 |

land are 0.89, 0.93, 0.98, 0.96, 0.96 and 0.89, respectively. The $R^2$s of KNN are the lowest among the four models and its $R^2$ of bare land is the lowest across all the land types. The prediction accuracy of the remaining models is at least 0.64. The MSEs of forest land, shrub grassland, artificial grassland, natural grassland, farmland, and bare land under the GBDT model are 0.12, 0.13, 2.63, 2.63, 22.34 and 30.19, respectively. The prediction accuracy of the remaining models is all lower than that of GBDT. KNN has the lowest MSE among the for models. Therefore, KNN is not suitable for simulating the sediment content in the

**Table 7  Validation of the predicted outcomes from the SY models of different control measures.**

| | NN model | | SVR model | | GBDT model | | KNN model | |
|---|---|---|---|---|---|---|---|---|
| | $R^2$ | MSE | $R^2$ | MSE | $R^2$ | MSE | $R^2$ | MSE |
| Arbor forest land | 0.91 | 0.95 | 0.71 | 0.10 | 0.89 | 0.72 | 0.84 | 2.76 |
| Shrub grass land | 0.76 | 2.09 | 0.85 | 2.14 | 0.93 | 0.13 | 0.91 | 1.66 |
| Natural grassland | 0.89 | 9.45 | 0.82 | 12.65 | 0.91 | 7.44 | 0.56 | 42.86 |
| Artificial grassland | 0.93 | 8.51 | 0.77 | 21.76 | 0.98 | 2.63 | 0.54 | 44.12 |
| Cultivated land | 0.94 | 65.45 | 0.96 | 25.01 | 0.96 | 22.34 | 0.77 | 134.45 |
| Bare land | 0.20 | 205.83 | 0.50 | 142.04 | 0.89 | 30.19 | 0.24 | 225.17 |

study zone and GBDT is the most accurate in predicting the sediment content among the four models.

## The structural equation model of influencing factors of RD and SY

Figure 5 is derived from the structural equation constructed based on factors affecting RD and the structural equation model (SEM). It was found that RD and soil erosion modulus increased as P, T, $I_{avg}$, $I_{30}$, and S increased. The influencing factors were positively correlated with RD and soil erosion modulus. Among them, P reached a noticeable level of $P < 0.01$ for RD and soil erosion modulus. T and $I_{avg}$ had a noticeable level of $P < 0.05$ for RD. There is also a noticeable connection between RD and SY. In summary, the structural equation models reveal that T, $I_{avg}$, $I_{30}$ and S are factors directly influencing RD, which has direct implications on SY. Therefore, T, $I_{avg}$, $I_{30}$ and S are factors that indirectly affect SY.

## DISCUSSION

### Effect of rainfall on RD and SY

Different prevention and ecological restoration measures are applied to areas with varying vegetation coverage, where different types of precipitation have varying impacts on water and soil loss. S is the key factor that leads to water and soil loss and it has complex relations with RD and SY (*Morbidelli et al., 2018*). According to *Wischmeier & Smith (1978)*, precipitation higher than 12.7 mm is erosive. *Jia et al. (2021)* analyzed the data of the runoff on the Loess Plateau and found that erosion happened in this region when the P is beyond 12 mm. *Zhang et al. (2022)* studied the erosive precipitation on the Loess Plateau based on the data of runoff in this region. They found that erosive precipitation was often had an $I_{30}$ of over 0.25 mm/min. The present study indicates that, under type I precipitation, the study area, where a variety of ecological restoration measures have been taken, undergoes the highest RD and amount of erosion. The section of the Kuyehe River has relatively frequent type I and II rainfalls, which of which are erosive precipitations and contribute significantly to the water and soil loss on the slopes in the region. This finding is consistent with the results of *Xu et al. (2017)*. This indicates that precipitation types characterized by short duration, high intensity and a large amount are more likely to to erosive. *Wei et al. (2025)* studied rainfall erosion of the Loess Plateau in Guizhou Province and found that frequent rainfall with small amounts was least erosive. The present study

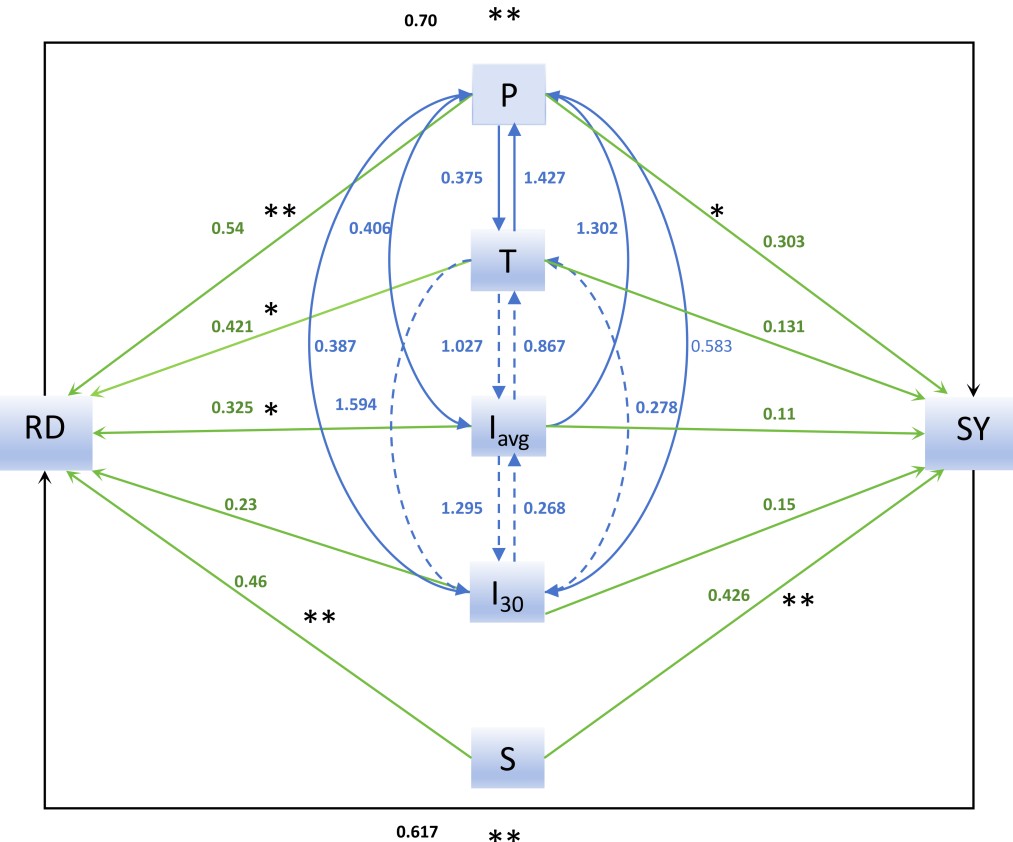

**Figure 5** **The structural equation model of influencing factors of runoff depth and sediment yield.**
Note: Figure 5 shows the structural equation model. The relationship between runoff depth, soil loss and their influencing factors is graphically displayed. The green arrows illustrate the influences of precipitation (P), T, $I_{avg}$ and $I_{30}$ and S on RD and SEM. The blue ones show the pairwise interaction among P, T, $I_{avg}$ and $I_{30}$. The black ones represent the interaction between RD and soil loss. (Asterisks indicate whether there is a significant difference between factors. One asterisk indicates $P < 0.05$, and two asterisks indicate $P < 0.01$.).

discovers that the erosion of the type III precipitation leads to the lowest RD and SY and best recharges the underground water of the study area. Under different ecological restoration measures, the three types of precipitation can be ranked as I > II > III in terms of RD and SY.

## Effects of ecological restoration measures on RD and SY

The study area is a sand-covered Pisha sandstone area. The sandstone is loose (*Morbidelli et al., 2018*) and is thus prone to erosion by surface runoff (*Fan, Qin & Che, 2024*). As a result, this area is characterized by sparse vegetation, exposed bedrock, thousands of gullies, and scattered sand dunes (*Zhu et al., 2024*). *Pierfranco et al. (2024)* studied the rainfall erosion on the Iberian Peninsula and pointed out that the eroding capacity of rainfall might vary significantly depending on the soil type and land use. Due to regular weeding, fertilization, and other agricultural activities, the surface has been exposed for a period of time. Human activities have destroyed the vegetation cover and the structure of

soil, whose particales became loose. This reduces the land's resistence to rainfall, and thus lowers its capacity in retaining water and soil. The rainfall and runoff have caused rills on the land surface. With the traceable development and continuous expansion of rills, RD and SY increase sharply, making the SY greater than that of bare land (*Luo et al., 2022*). Compared with bare land and cultivated land, grassland, shrub grassland, and arbor forest land have higher vegetation coverage, hence higher water and soil retaining capacity and seepage reduction effects (*Wang et al., 2022*; *Shi et al., 2024*). Compared with a single-layer community, a multi-layer community is more resistant to water erosion (*Liu et al., 2023*). This is similar to the findings of *Zhu et al. (2024)*, who studied the soil erosion effects of various ecological restoration measures in the loess hilly and gully region of China. Compared to grasslands, arbor and shrub forests not only provide canopy interception and buffering effects on rainfall (*Casermeiro et al., 2004*), but also enhance soil stabilization through the root systems of their vegetation (*Zhu et al., 2024*; *Wang et al., 2024*). Therefore, the SY on the slope covered by trees and shrubs is less than that of grassland.

## Effects of slope gradient on RD and SY

As an important factor influencing the erosion of slope surface, S is commonly incorporated into equations used to predict erosion (*Liu et al., 2024a*). *Renard & United S Agricultural Research Service (1997)* pointed out that there is a positive correlation between S and the volume of soil erosion. S affects the hydrodynamic forces of water, the stability of the slope surface, and consequently the effectiveness of water and soil conservation measures (*Chen et al., 2022*). S also affects water and soil conservation under different precipitation types on lands used for different purposes. When the S and precipitation are low, bare land do not differ much in water retention capacity from lands for other uses. When the S reaches $15°$, however, the runoff on bare land are significantly deeper than that in grassland, arbor forest land and shrub grassland. Grasslands are characterized by good infiltration, while proper cultivation and management of forests and lands with shrubs and grass can lead to the shrub cluster effect, which stabilizes the soil and promotes infiltration, thereby reducing RD. Therefore, we can observe that S directly affects the erosivity of runoff, and soil erosion deteriorate as S increases.

## Identifying suitable machine learning models

The models of RD and SY are developed based on the data of subsequent rainfall under different ecological restoration measures. GBDT has higher accuracy in predicting RD than the remaining three models with similar complexity and is thus the most suitable for the study area. The MSE ranges from 0.09 to 3.88, indicating that its accuracy in predicting RD is relatively stable. When it comes to the prediction of SY, GBDT is also the most suitable model, the MSE ranging from 0.12 to 30.19. According to *Wang, Qin & Yu (2007)*, despite the significant positive correlation between them, RD and SY might be affected differently by the same factor. The variation can be attributed to the model's inability to process certain figure in the data set, which leads to the prediction error. The present study finds that GBDT has the highest prediction accuracy, compared to the other three models. This finding coincides with *Arabameri et al. (2020)*. *Liu, Fan & Wang (2024)* studied the runoff

erosion in China's northeast using the models of Random Forest (RF), Convolutional Neural Network (CNN) and Transformer and pointed out that Transformer was the most applicable among the three. This may be due to the variation in performance metrics between the training and validation sets, which indicates overfitting. *Bag et al. (2022)* studied Sobha Basin in West Bengal, India, with a combination of geographic information system (GIS) spatial analysis techniques and machine learning models and found that RF was significantly more accurate than other models in predicting soil erosion (Area Under the Curve (AUC) = 0.97). This indicates that the selection of models is dependent on study areas, training sets and validation sets.

### Main factors influencing runoff depth (RD) and sediment yield (SY)

The structural equation model indicates that RD is a factor directly influencing soil loss and that there is a strong correlation between the RD and soil loss (*Chen et al., 2022*). Among the factors in discussion, P, T, $I_{avg}$ and S have significant impacts on RD, while P and S have significant influences on soil loss. *Ran, Wang & Gao (2019)* points out that rainfall patterns and characteristics are controlling factors in runoff and soil erosion processes. This study found that for short-duration rainfall events, the flow erosivity and erosion amounts exhibit a trend which first increases with slope gradient, and then decreases. *Wu et al. (2017)* also found in his study that rainfall intensity has great influence on the slope effect trends. Thereby, we can see that P and S directly influence RD and soil loss.

## CONCLUSIONS

The present study is conducted on the data collected from long-term observations in standardized runoff plots where a variety of water and soil conservation and vegetation restoration have been taken. The results showed that the rainfall of the temple soil and water conservation monitoring station was mainly concentrated from July to September. Among them, the rainfall events with short duration, medium rainfall, and medium rainfall intensity were the most frequent. All the ecological restoration measures pointed out that the rainfall events with medium duration, heavy rainfall, and heavy rainfall intensity had the most serious runoff and sediment.

Compared with the control plot (bare land), ecological restoration measures can noticeably reduce runoff and SY. The influence of ecological restoration measures on RD and SY varied with ecological restoration measures. Among them, arbor forest and shrub grassland were the best ecological restoration measures for controlling soil loss because of their canopy structure. Tree- and shrub-forests are effective in preventing and control soil erosion. Using machine learning models, the researchers found that GBDT performed best in simulating soil erosion in the study area. The structural equation model showed that there is a significant correlation between RD and SEM. T, $I_{avg}$, $I_{30}$ and S are factors that directly affect SY. Since RD directly impacts SY, it can be said that T, $I_{avg}$, $I_{30}$ and S indirectly influence SY. Our research findings can provide theoretical guidance and technical support for the prevention and control of soil erosion under different slope management measures

in the sand-covered soft rock area of the middle reaches of the Yellow River, thereby contributing substantially to the ecological restoration in the Yellow River Basin.

### Funding
This research was supported by the National Key Research and Development Program of China (No. 2023YFF1305104), National Natural Science Foundation project (42307463), Inner Mongolia Autonomous Region "Science and Technology to Prosper Mongolia" Action Key Project (2022EEDSKJXM003), Inner Mongolia Autonomous Region Science and Technology Plan Project (2021GG0052), "The integrated benefit evaluation of soil and water conservation comprehensive improvement technology in the key soil erosion area ecosystem (Xiheidai watershed) of Ordos water conservancy science and technology project", Double first-class construction funds-subject construction quality improvement project. The funders had no role in study design, data collection and analysis, decision to publish, or preparation of the manuscript.

### Grant Disclosures
The following grant information was disclosed by the authors:
National Key Research and Development Program of China: 2023YFF1305104.
National Natural Science Foundation project: 42307463.
Action Key Project: 2022EEDSKJXM003.
Inner Mongolia Autonomous Region Science and Technology Plan Project: 2021GG0052.

### Competing Interests
The authors declare there are no competing interests.

### Author Contributions
- Xin Wang conceived and designed the experiments, performed the experiments, analyzed the data, prepared figures and/or tables, and approved the final draft.
- Zhenqi Yang conceived and designed the experiments, authored or reviewed drafts of the article, and approved the final draft.
- Jianying Guo conceived and designed the experiments, authored or reviewed drafts of the article, and approved the final draft.
- Fucang Qin analyzed the data, authored or reviewed drafts of the article, and approved the final draft.
- Xinyu Liu performed the experiments, authored or reviewed drafts of the article, and approved the final draft.
- Dong Fan performed the experiments, prepared figures and/or tables, and approved the final draft.

### Data Availability
The raw measurements are available in the Supplementary File.

## Supplemental Information

Supplemental information for this article can be found online at http://dx.doi.org/10.7717/peerj.20040#supplemental-information.

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
