# Peer review of "Driving factors of runoff and sediment yield effects of various slope ecological restoration measures in the middle reaches of the Yellow River"

_PeerJ, doi:10.7717/peerj.20040_

## Round 0.1 · original submission · Major Revisions

Based on the comments from two anonymous referees, your manuscript requires major revisions before it can be accepted. In addition, the language needs to be polished by native English speakers.

**Language Note:** We note that you have already obtained editing services from PeerJ. However, PeerJ copyeditors are not subject specialists and only edit the language. It is your responsibility to ensure the text conveys your intended message. It might be helpful to enlist the help of a colleague with relevant subject knowledge.

Reviewer 1 ·

Basic reporting

- The language is often repetitive, not always technically precise, and sometimes lacks fluidity. Some phrasing is ambiguous or overly generic. A careful professional English editing pass is strongly advised to enhance clarity, precision, and flow. Sentences should be more direct.

- Relevant literature is cited, but mostly national or regional Chinese studies. There is little connection to broader, international soil erosion research. I suggest to expand the literature review to include global soil erosion studies and international examples of watershed management. For example the authors can include 1. DOI:10.1016/j.jhydrol.2025.132666, 2. DOI: 10.1016/j.jhydrol.2024.130778

- The structure of the paper could be improved. A distinct 'Conclusion' section is missing; the overall text would benefit from a more logical and coherent organization.

- While the authors collect extensive observational data, the overall methodological rigor is insufficient for a study that claims to provide a "scientific basis" for runoff and sediment yield responses. The statistical analysis, mainly based on basic regression and path analysis, is too superficial to capture the complexity of soil erosion processes. Incorporating more robust, multivariate or nonlinear modeling approaches (e.g., machine learning models, structural equation modeling) could have greatly improved the explanatory power and generalizability of the results.

Experimental design

- The manuscript states that it aims to address a gap in understanding the mechanisms of runoff and sediment yield under different vegetation control measures. However, the research question is not explicitly formulated as a clear, focused statement or hypothesis. Instead, the manuscript presents a broad research aim without clearly structured research questions or specific hypotheses that guide the study. It is recommended to explicitly define 1–2 clear research questions or hypotheses at the end of the Introduction section to sharpen the focus of the study and strengthen the logical flow.

- The study relies on six years of field data collection, which indicates a serious long-term commitment to high technical standards. However, the data analysis remains relatively basic (mostly regression and
descriptive statistics), and does not fully exploit the richness of the collected data.

- The study relies on six years of field data collection, which indicates a serious long-term commitment to high technical standards. However, the data analysis remains relatively basic (mostly regression and descriptive statistics), and does not fully exploit the richness of the collected data, due to the lack of modelling analysis.

Validity of the findings

- The manuscript lacks a discussion on how the findings contribute to advancing the field or why this study is novel. There is no comparison to existing literature in terms of how this study moves the field forward. The authors should explicitly state the novelty and impact of their research. A section discussing the unique contributions of their work compared to existing studies, and how it advances knowledge or informs future research directions, is required.

-

Reviewer 2 ·

Basic reporting

General Comments:
1. The article focuses on the main causes of slope erosion in the Yellow River Basin. It is essentially a case study. The research problems discussed in the article are well-known and broadly addressed in the global scientific literature. It seems that the authors have insufficiently reviewed the relevant literature, which is evident both in the Introduction section and in the Discussion section. The authors cite only a few studies, mostly from China.
In the Introduction section, it is necessary to clearly state what is already known about the influence of various factors (e.g., slope gradient, land use, topography, climatic conditions) on slope sediment erosion. These factors should be listed, and references should be made to specific studies from different regions of the world that examine the influence of each factor. Against this background, the current state of knowledge regarding the study area in the middle reaches of the Yellow River should be presented.
The authors refer to the strong impact of topography (line 53), but this factor is not actually considered in the study. Please explain this discrepancy or include topography in the analysis.
The exclusion of geological substrate and soil types from the analysis also appears questionable—please clarify this. Why were only rainfall, slope gradient, and land use included as erosion-driving factors? What was the rationale behind selecting only these factors?
In the Discussion, the results should be compared with those of similar studies conducted in other parts of the world, pointing out similarities and differences.
2. The manuscript requires improvement in terms of English language quality, as it contains linguistic errors, repetitions, and unclear expressions.
3. The reference list is disorganized and poorly formatted. The literature is cited in various styles, and alphabetical order is not maintained in several places. Please revise it using a consistent citation style compliant with the journal's guidelines.
4. All figures lack captions, making them difficult to understand.
* * *
Specific Comments:
1. The Introduction section is missing a title. It appears to begin at line 34, but there is no heading.
2. The text in the Introduction starting with: “The Kuye River Basin has the most serious Sy problem…” (line 41) and ending with “…transport modulus is 1,157 tons / km² (Wang et al., 2023)” (line 51) should be moved to the section 1.1 Overview of the study area.
3. The description of the study area lacks detail, particularly regarding topography. Soils are not the same as topography (line 75)—please describe topography and soils separately, as both have a significant impact on erosion processes.
4. Figure 1 is too general; it shows the entire Yellow River Basin. The text refers to specific geographic names within the study area, such as “Subulga Town of Yijinhuoluo Banner, Ordos” and “Shenmu County”, which are not marked on the map.
Moreover, it is unclear whether the study is conducted in the Kuye River Basin (as in lines 66–67) or the Miaochuan Basin (line 76). It is difficult to visualize the study area based on such vague information and a general map.
Please present the studied catchment area together with the experimental plots in a separate figure (or combined with Figure 2).
5. In Figure 1, "Kuye River" is written with a lowercase letter—please correct it.
6. What do the two photos in Figure 1 show? Why were they included? If the authors decide to retain them, please label them appropriately and improve their quality.
7. The description of methods should appear in a separate main section titled 2. Materials & Methods, not as subsection 1.2.
8. Please remove section 1.2 Experimental design.
o The content from lines 84–90 should be placed in section 1.1, Overview of the study area.
o The text from lines 90–97 should be moved to the beginning of section 2. Materials & Methods (as it describes the setup of the experimental plots).
9. The section is currently titled 2. Results and analysis should be renamed to 3. Results.
10. The content between lines 125–129 does not belong in the section on precipitation characteristics and repeats information already provided in section 1.1. Please remove it.
11. The paragraph:
“Table 3 shows that the rainfall erosivity was type I > type II > type III. The average rainfall erosivity of type II was the largest, which is 0.6 times and 3.1 times that of type I and type III, respectively, which is very easy to cause soil erosion” (lines 137–140) is unclear. These observations are not reflected in the table, and types I and II rainfall appear combined. Please revise the table and clarify the statement.
12. The resolution of Figure 4 is very low, making it almost unreadable.
13. How are the different rainfall types shown in Figure 4? This is suggested in line 146—please clarify.
14. Figure 5 is unclear—there is no caption or explanation.
15. The content in lines 199–207 in the Discussion repeats earlier information. Please remove it.
16. Tables 4 and 5: What do the asterisks next to some values mean?
17. Table 8: In the caption, “sediment” should be capitalized.
18. The section “Regression analysis of factors affecting runoff and sediment” lacks interpretation. What do the results mean?
19. In the sentence:
“Other studies have pointed out that rainfall types with short duration, strong rainfall, and heavy rainfall are more likely to cause erosive rainfall” (lines 213–215),
There is no reference—please cite the studies you are referring to.
20. The sentences:
“The runoff and sediment response of each treatment measure under various rainfall patterns was basically the same. The order of Rd and Sy was type I > type II > type III” (lines 215–216)
They are too vague. Please rewrite for clarity.
21. The sentence: “Therefore, its S Sy is lower than that of grassland” (line 246) needs to be corrected.
22. The paragraph in lines 247–258 discusses the influence of slope gradient, not land cover, and does not belong in that section. Consider creating a new section:
“Effects of slope gradient on Rd and Sy”.
23. The section “The influencing factors of Rd and Sy” should be renamed:
“Main factors influencing runoff depth (Rd) and sediment yield (Sy)”.
24. The sentence in lines 260–261 appears unnecessary.
25. These four sentences:
“For the Rd and Sy model of various control measures, the main driving factors of the multiple regression model and path analysis included rainfall and S. Regardless of the kind of control measures, the regression model and path analysis of Rd and Sy all had rainfall and S as explanatory factors. Rainfall and S had a direct positive impact on Rd and Sy. Rainfall and S are not only the key factors affecting Rd, but also the driving factors leading to change in Sy (Kooch et al., 2023)” (lines 261–266)
are repetitive. Please shorten this section to avoid redundancy.
26. In the Abstract, the section:
“Therefore, addressing this issue is the key to sediment control in the Yellow River Basin. However, erosion and sediment yield (Sy) patterns, as well as the influencing factors of various slope (S) control measures in this area, remain unclear. This limits our understanding of the Sy mechanism and hinders the effectiveness of soil erosion control. It is important to investigate the law of erosion and Sy across various control measures” (lines 17–21)
contains many repetitions. Please revise and shorten this part.
27. In the sentence:
“We analyzed the variation in runoff and Sy under different control measures, and the results will provide technical guidance for the ecological restoration and improvement of various control measures in the typical watershed of the middle reaches of the Yellow River” (lines 31–33),
the phrase “We analyzed the variation in runoff and Sy under different control measures” is repetitive and should be removed.

Experimental design

1. The research objective stated in the Introduction (lines 69–71) is phrased differently than in the Conclusions (lines 271–273). Please rewrite the objective clearly and consistently. It should appear in the Introduction in a separate paragraph, and the authors should explicitly state the novelty and relevance of their research—what gap is being addressed, and what is the practical importance?

Validity of the findings

1. There is no Conclusions section. The section titled "4 Results" should be renamed to "5 Conclusions."
2. The Conclusions should include recommendations for sediment control and watershed management in the Yellow River Basin based on the study's findings.

---

## Round 0.2 · Minor Revisions

The manuscript has been significantly improved. However, some minor comments are still requried to address before it can be accept based on the second reviewer.

Reviewer 1 ·

Basic reporting

The authors have addressed all of my comments and suggestions. I believe the paper has significantly improved and is now ready for publication. The only point I would like to highlight is that, in some of the newly added references, there appears to be some confusion between the authors' first and last names. I kindly ask that this be checked and corrected accordingly.

Experimental design

See my previous comment

Validity of the findings

See my previous comment

Reviewer 2 ·

Basic reporting

The manuscript has been significantly improved. The authors have responded thoroughly to each comment, for which I thank them. The article's structure is now appropriate, and the overall use of English has improved. However, the manuscript still requires further language polishing. Some sections remain incorrectly written in English (detailed below), and I strongly recommend a full professional language revision before publication.
The subsection "4.5. Challenges in Runoff and Sediment Research in the Middle Reaches of the Yellow River" fits better in the Introduction, where it could justify the research relevance. Please consider moving this section to the Introduction.
The division into sections and subsections has been corrected and is now appropriate. However, there is still a lack of paragraph structure within the sections. Please revise the text to include paragraph breaks to improve readability and logical flow.
The figure captions are still missing in the PDF version of the manuscript; they are absent both in the main text and beneath the figures.

Experimental design

OK

Validity of the findings

OK

Additional comments

1. Introduce abbreviations such as runoff depth (RD) and sediment yield (SY) at their first occurrence (line 46).
2. Line 50 – Please clarify what is meant by “regulatory measures.” If this refers to engineering structures, land maintenance works, or specific land-use practices, this should be clearly stated. The term “regulatory measures” is too vague on its own. Similarly, in lines 62–65, “control measures” are discussed without explanation. Please specify what types of interventions are included in these categories.
3. Line 50 – Please define the abbreviation I30.
4. Line 57–58 – What does S represent in the sentence: “S and slope length are two factors related to the implications of landform on soil erosion”? Please clarify the meaning of S.
4. Lines 81–82 – The sentence is grammatically incorrect. Suggested revision:
“The upper reaches of the Kuye River Basin are characterized by a landscape covered with sand dunes and shifting sands, while the lower reaches are located in a loess hilly and gully region.”
5. Lines 91–92 – Suggested improvement:
“Situated at the northern edge of the Mu Us Sandy Land, the area is characterized by sand-covered hills and gullies.”
6. Line 95 – The sentence is unclear and grammatically incorrect:
“The typical watershed was selected to be the soil and water conservation monitoring network site of Yijinhuoluo Banner, Ordos Hetong Temple, which was built in 2016 (Fig. 2).”
Please rewrite for clarity. It’s not clear what was built and what exactly is meant.
7. Figures 1 and 2 include the same map. This must be corrected. I suggest removing the Miaochuan Basin map from Figure 1 and keeping it only in Figure 2 (the opposite of how it is now).
8. Figure 1 caption – Please explain what is shown in parts (a) and (b).
9. Line 112 – Define the abbreviation S.
10. Line 113 – What does "indicator rainfall (P)" mean? Does it refer to rainfall amount? Please clarify.
12. Line 119 – Please introduce the formulas presented in lines 120–121. Add a sentence such as:
“Runoff depth and sediment yield were calculated using the following formulas:”
13. Lines 152–153 – The sentence:
“type I, type II, and type III rainfall accounted for 80.6%, 18.6%, and 0.8%, respectively”
is grammatically incorrect and starts with a lowercase letter. Please revise.
14. Lines 153–154 – Suggested correction:
“Table 2 indicates that the average rainfall intensity decreased in the order: type I > type II > type III.”
However, the next sentence says that type II had the highest intensity, which contradicts the previous sentence. Please check the data and correct accordingly.
15. Lines 164–166 – This section is unclear and poorly written in English. Also, it suddenly mentions “type III erosion,” while earlier the classification was based on rainfall types. Please clarify.
16. Lines 173–181 – This section is poorly structured and unclear. Please revise for logical flow and clarity.
17. Line 205 – Define the abbreviation SEM at first mention.
18. Tables 5 and 6 – These tables have the same title. It is unclear what new information Table 6 provides compared to Table 5. Are the differences in data intentional? Please explain and differentiate.
19. Lines 219–222 – This passage appears both in the main text and below Figure 5. Please remove it from the main text.
20. Lines 239–241, 253–254, 264–266, 272–275 – These sections are unclear, contain grammatical errors, and sometimes repeat ideas. Please revise for clarity and conciseness.
21. Lines 289–301 – This section addresses both research objectives, although the heading suggests only one: identifying key factors affecting erosion. Please split the text and add a separate subsection discussing the second objective – identifying suitable machine learning models.

---

## Round 0.3 · accepted · Accept

The authors have addressed all the comments from the reviewers. The current manuscript can be accepted.